# Heterologous Expression of a *Glycine soja* C2H2 Zinc Finger Gene Improves Aluminum Tolerance in *Arabidopsis*

**DOI:** 10.3390/ijms21082754

**Published:** 2020-04-15

**Authors:** Yuan-Tai Liu, Qi-Han Shi, He-Jie Cao, Qi-Bin Ma, Hai Nian, Xiu-Xiang Zhang

**Affiliations:** 1The State Key Laboratory for Conservation and Utilization of Subtropical Agro-bioresources, South China Agricultural University, Guangzhou 510642, China; liuyuantai@stu.scau.edu.cn (Y.-T.L.); shiqihan@stu.scau.edu.cn (Q.-H.S.); 2651465420@stu.scau.edu.cn (H.-J.C.); maqibin@scau.edu.cn (Q.-B.M.); 2The Guangdong Provincial Laboratory of Lingnan Modern Agricultural Science and Technology, South China Agricultural University, Guangzhou 510642, China; 3The Key Laboratory of Plant Molecular Breeding of Guangdong Province, College of Agriculture, South China Agricultural University, Guangzhou 510642, China; 4The Guangdong Subcenter of National Center for Soybean Improvement, College of Agriculture, South China Agricultural University, Guangzhou, 510642, China

**Keywords:** *Glycine soja*, C2H2 Zinc finger protein, *GsGIS3*, *Arabidopsis*, Al tolerance

## Abstract

Aluminum (Al) toxicity limits plant growth and has a major impact on the agricultural productivity in acidic soils. The zinc-finger protein (ZFP) family plays multiple roles in plant development and abiotic stresses. Although previous reports have confirmed the function of these genes, their transcriptional mechanisms in wild soybean (*Glycine soja*) are unclear. In this study, *GsGIS3* was isolated from Al-tolerant wild soybean gene expression profiles to be functionally characterized in *Arabidopsis*. Laser confocal microscopic observations demonstrated that GsGIS3 is a nuclear protein, containing one C2H2 zinc-finger structure. Our results show that the expression of *GsGIS3* was of a much higher level in the stem than in the leaf and root and was upregulated under AlCl_3_, NaCl or GA3 treatment. Compared to the control, overexpression of *GsGIS3* in *Arabidopsis* improved Al tolerance in transgenic lines with more root growth, higher proline and lower Malondialdehyde (MDA) accumulation under concentrations of AlCl_3_. Analysis of hematoxylin staining indicated that *GsGIS3* enhanced the resistance of transgenic plants to Al toxicity by reducing Al accumulation in *Arabidopsis* roots. Moreover, *GsGIS3* expression in *Arabidopsis* enhanced the expression of Al-tolerance-related genes. Taken together, our findings indicate that *GsGIS3*, as a C2H2 ZFP, may enhance tolerance to Al toxicity through positive regulation of Al-tolerance-related genes.

## 1. Introduction

Crop growth is restricted on acid soils, which accounts for about 50% of global arable lands [1,2]. When the soil has a pH below 5, Al mainly exists as Al^3+^, which forms an Al complex in the soil [3,4]. Al^3+^ can inhibit root development by damaging cell walls [5] and the structure of cytoskeletons [6], affecting the plasma membrane structure [7,8,9] as well as impeding the stress of signaling pathways [10]. Consequently, nutrient uptake is inhibited, and root cell death is induced, which leads to significant reductions in crop yield [11,12]. Therefore, Al toxicity has been widely acknowledged as a global problem for plant growth and crop yield on acidic soil [11,12].

In the past few decades, many studies have clarified the mechanisms of plant tolerance to Al toxicity [13,14,15]. Up to now, two main mechanisms of Al resistance, Al exclusion and Al tolerance, have been investigated in most plant species. The external exclusion mechanism is that the root cells exclude Al^3+^ from the protoplasts by some ways, while the internal tolerance mechanism is to chelate Al^3+^ in the cell sap or isolate it to vacuoles [1]. Some plants chelate and detoxify Al^3+^ by secreting organic acids, such as malate, citrate and oxalate [16,17]. The gene Al-activated malate transporter 1 (*ALMT1*), which encodes an anion transporter/channel, was first identified in wheat. *ALMT1* is responsible for Al-activated secretion of malate in plant roots. Previous studies showed that plants could release both malate and citrate in response to Al stress, with malate being essential for Al detoxification [11,18]. Al toxicity is mainly concentrated to the root apex, where phytohormones such as gibberellin, abscisic acid (ABA), and cytokinin are synthesized [19]. A previous study has reported that NO induced the synthesis of auxin and gibberellin to promote the degradation of the GA signal transduction inhibitory protein (DELLA proteins), which improved the plant’s tolerance to aluminum toxicity [19]. Thus, it can be seen that some phytohormones are involved in the Al-stress response. Plants also activate antioxidant defense to mitigate oxidative damage caused by aluminum toxicity; the defense systems include superoxide dismutase (SOD), peroxidase (POD), catalase (CAT), ascorbate peroxidase (APX), glutathione (GSH) and proline. Plants can remove reactive oxygen species (ROS) produced by aluminum stress in accordance with these complex enzyme systems; thus, enhancing plant antioxidant capacity can improve plant resistance to aluminum toxicity. Previous reports indicated that proline content in plants increases under stress, and a large amount of proline can alleviate the damage caused by adversity stress [20]. MDA content is an important parameter to reflect the potential antioxidant capacity of plants [21]. Proline and MDA can be used as physiological indicators for evaluating the resistance of plants to abiotic stress.

The zinc-finger protein (ZFP) family is one of the largest family of transcription factors in plants [22]. According to the number and position of the cysteine and histidine residues, zinc-finger proteins are divided into C2H2, C4, C6, C4HC3, C2HC, C2HC5, and combined types. Among them, C2H2 ZFP is a eukaryotic transcription factor, which contains 1–4 zinc finger structures, and, in plants, it has a highly conserved QALGGH motif, which plays an important role in abiotic stress [23]. Since the first C2H2 ZFP gene *EPF1* was isolated from petunia [24], more C2H2 ZFP family genes have been found in plants and widely reported to be involved in responses to biotic and abiotic stresses [25]. In *Arabidopsis*, constitutive expression of *AtZAT7* or *AtZAT10* improved salt tolerance [25,26]. Overexpression of *ZAT18* enhanced drought tolerance of *Arabidopsis* [27], whereas *AtZAT6* could regulate biotic and abiotic stress tolerance through the Salicylic Acid (SA) activation pathway [28]. *AtSIZ1* increased the tolerance to salinity by maintaining ionic homeostasis and osmotic equilibrium in *Arabidopsis* [29]. In rice, *OsZFP179* participated in response to salt stress [30]. The C2H2 ZFP gene, *Al resistance transcription factor 1 (ART1)*, mainly regulated the expressions of Al-tolerance-related genes in rice [31]. *OsZFP252* could respond to salt and drought stress, whereas its overexpression enhanced proline and soluble sugar contents in rice plants as well as the expression of drought stress-related response genes was significantly upregulated [32]. In addition, *OsZFP245* was induced by drought and cold and the over-expressed transgenic plants showed resistance to low temperature and drought [33]. Similar to rice and *Arabidopsis*, many ZFP genes that were involved in abiotic stress have been reported in other species. For instance, *GhSTOP1* enhanced the resistance to Al and proton stress in transgenic cotton plants by delaying the lateral root initiation process. Heterologous expression of *Zoysia japonica* gene, *ZjZFN1*, improved salt tolerance in *Arabidopsis* [34]. A wheat gene *TaZAT8* played essential roles in regulating tolerance to Pi deprivation through mediating P acquisition, ROS homeostasis and root system development [35]. Among the ZFP genes in soybean, over-expression of *GsZFP1* improved the tolerance to drought in *Arabidopsis*, while *GmZFP3* negatively regulated its drought tolerance [23]. Overexpression of *GmZF1* made the transgenic *Arabidopsis* more adaptable to cold stress [36]. *GmSTOP1* members have been found to play a critical role in H^+^ and Al toxicity through transcriptional regulation of H^+^ tolerance and Al tolerance genes [37].

Soybean is one of the most important oil and protein crops in the world, providing approximately 28% of the edible oil and 67% of the plant protein sources for human beings [38]. Wild soybean is a relative ancestor of soybean. In the process of domestication and improvement of soybeans, the genetic diversity of soybeans has declined sharply. The annual wild soybean has a large amount of genetic diversity and has preserved the excellent qualities lost in soybean. Therefore, via our previous work, we constructed Al-tolerant wild soybean gene expression profiles using the collected wild soybean resource in order to study the mechanism of resistance to acid aluminum stress. Although many studies have identified the functions of ZFP family members in abiotic stress and growth development in other plant species, the potential transcriptional regulation mechanism of Al tolerance in soybean is still unknown. In the current study, we cloned a C2H2 zinc-finger protein, *GsGIS3*, from the wild soybean BW69 line (Al-resistant line of *Glycine Soja*), aiming to analyze its role in Al stress tolerance and investigate its potential transcriptional regulation mechanism.

## 2. Results

### 2.1. Isolation and Bioinformatics Analysis of GsGIS3

A C2H2 ZFP gene, which was named *GsGIS3*, induced by acidic Al stress was isolated on the basis of Al-tolerant wild soybean gene expression profiles. The *GsGIS3* cDNA sequence was obtained from the NCBI database under gene locus LOC100811796 and protein accession number of XP_003536676.1. The gene *GIS3*, located on soybean chromosome 10, the full-length genome sequence of *GsGIS3* included 1 exon with an open reading frame (ORF) 852 bp in length. The predicted GsGIS3 protein comprises 283 amino acids (AA) (Figure 1A). The GsGIS3 protein contained one typical zinc finger structure and a QALGGH motif (Figure 1A). Its theoretical isoelectric point (pI) was 6.86, and its molecular weight (MW) was 29.66 KD. Phylogenetic analysis showed that GsGIS3 was most closely similar to the zinc finger protein GIS3 (GLABROUS INFLORESCENCE STEMS3) from *Glycine max* (Figure 1B).

### 2.2. Subcellular Localization of GsGIS3

We predicted by using the online website that GsGIS3 protein was localized in the nucleus. To verify the exact subcellular localization of GsGIS3, we constructed the fusion expression vector pCAMBIA1302-GsGIS3-GFP and conducted transient expression in *N. benthamiana* leaf cells. As determined by the laser confocal microscopic observations, pCAMBIA1302-GsGIS3-GFP protein accumulated mainly in the nucleus with strong signals of green fluorescence. In contrast, GFP alone was distributed evenly throughout all parts of the cell including the nucleus and cytoplasm (Figure 2). The results show that GsGIS3 was localized within the nucleus, which suggests that the GsGIS3 protein may act as a transcription factor.

### 2.3. Expression Patterns of GsGIS3

The quantitative real time polymerase chain reaction (qRT-PCR) was used to investigate the expression patterns of *GsGIS3*. As the reports showed, *GsGIS3* was mainly expressed in the root and stem. However, *GsGIS3* was expressed to a lesser degree in the leaf (Figure 3A). The expression level of *GsGIS3* was higher in 2–4-cm root segments than in 0–2- and 4–6-cm root segments. In addition, *GsGIS3* expression was considerably elevated in all tissues with Al treatment (Figure 3A,B). The results suggest that GsGIS3 protein may function as a C2H2 ZFP transcription factor in root and stem development in soybean. Compared to the control, different Al treatment concentrations could all significantly induce the upregulation of GsGIS3 expression, while the expression improved up to the highest level of five-fold at the treatment of 50 μM AlCl_3_ compared with those detected under control condition (Figure 3C), and the expression level decreased to 2.5-fold at the treatment of 100 μM AlCl_3_ compared with those detected under control condition. In addition, we analyzed the temporal expression pattern of *GsGIS3* under acidic Al treatment over a period of 24 h; the experimental results indicated that *GsGIS3* was quickly induced by Al treatment after a short time period and increased up to the highest expression level of 4.5-fold at 6 h (Figure 3D). To investigate the expression patterns of *GsGIS3* response to other abiotic stresses, we analyzed the transcript abundance of *GsGIS3* under NaCl, GA3, 6-BA and ABA treatments. Similarly, the transcription of *GsGIS3* could be enhanced by NaCl and GA3 treatments. Under 200 mM NaCl treatment, the expression of *GsGIS3* was increased rapidly up to the highest level of 9.3-fold after 6 h and then reduced to a lower level under the 12–24 h treatments (Figure 3E). For 100 μM GA3 treatment, *GsGIS3* was also induced rapidly and similarly reached the highest expression level of 10.9-fold at 4 h, then the expression of *GsGIS3* dropped a little and maintained at that level for a long time (Figure 3F). However, we found that *GsGIS3* was not subjected to 6-BA and ABA treatment in this study (Figure 3G,H). The results suggest that *GsGIS3* might play a potential role in Al stress or GAs-mediated pathways.

### 2.4. Overexpression of GsGIS3 Enhanced Plant Tolerance to Al toxicity

To explore the tolerance of *GsGIS3* transgenic lines to Al stress, we identified the transcription levels of five homozygous transgenic lines and selected three lines with higher expression levels for subsequent experiments. *Arabidopsis* seedlings were treated at 0, 50 and 100 μM AlCl_3_ for seven days. Under the control growth conditions, no obvious difference was found between wild-type and transgenic *Arabidopsis*. Under treatment of 100 μM AlCl_3_, all *Arabidopsis* roots were inhibited in varying degrees compared to the control; the root length of wild-type plants was 3.4 cm, while the root lengths of *GsGIS3* transgenic lines ranged from 4 to 4.5 cm (Figure 4A,B). The transgenic lines showed better growth than wild-type plants in Al stress. Furthermore, under the control conditions, no difference was found in both wild-type plants and *GsGIS3* transgenic lines. The contents of proline and MDA were improved in Al stress. However, under 100 mM AlCl_3_ treatment, proline contents in WT plants increased to 23 μg·g^−1^ of fresh weight while the proline contents in transgenic lines ranged from 43 to 56 μg·g^−1^ of fresh weight in transgenic lines, (Figure 4C). In addition, under 100 mM AlCl_3_ treatment, MDA accumulation in WT plants improved to 0.019 μmol.g-1 of fresh weight, while MDA contents in *GsGIS3* transgenic lines increased to the range of 0.010–0.013 mmol·g^−1^ of fresh weight in Al stress (Figure 4D). MDA in wild-type was significantly higher than that of transgenic lines.

To further verify the tolerance of *GsGIS3* to Al stress, seven-days-old transgenic lines and WT seedings were transformed to 1/30 Hoagland nutrient solution (without NH_4_H_2_PO_4_ and plus 1 mM CaCl_2_ and 2 μM AlCl_3_) for 21 days. Under the control growth conditions, no obvious difference was found between wild-type and transgenic *Arabidopsis*. The total root lengths of transgenic lines and WT were about 130 cm, and the surface area of that was about 12 cm^2^. However, under AlCl_3_ treatment, all observed plants performed delayed growth and depressed root elongation; the total root length of the WT was 29 cm, while the total root lengths of *GsGIS3* transgenic lines ranged from 43 to 47 cm. The SA of the WT was 2.2 cm^2^, while the SA of *GsGIS3* transgenic lines ranged from 3.9 to 4.4 cm^2^. Less damage was found in transgenic lines compared to WT plants, which showed a higher total root length and total root surface area (Figure 5A–C). The results of phenotypic identification indicate that overexpression of the *GsGIS3* gene enhanced the tolerance of transgenic Arabidopsis to Al stress.

### 2.5. Overexpression of GsGIS3 Reduces the Hematoxylin Staining Degree of the Hairy Roots

To further verify the function of *GsGIS3* in Al tolerance, homologous expression of *GsGIS3* in soybean hairy roots was done using Hematoxylin staining. It was found that the staining of soybean hairy roots with the overexpression of *GsGIS3* was shallower and that it was deeper in RNAi soybean hairy roots compared to control (Figure 6), indicating that overexpression of *GsGIS3* improves the tolerance of the hairy roots to Al stress by reducing Al accumulation in soybean roots.

### 2.6. Expression Patterns of Al Stress/GA3-related Genes Regulated by GsGIS3

To further explore the potential transcriptional mechanism, we used qRT-PCR to assess the differential expression of these genes between *GsGIS3* transgenic lines and WT plants. Two related Al tolerance genes *AtALMT1* and *AtALS3* and three GA-related genes including Gibberellin synthetase (GA3-oxidase) *AtGA3OX1*, Gibberellin synthesis receptor (Gibberellin insensitive dwarf1) *AtGID1*, and Gibberellin inactivation enzyme (2-oxoglutarate-dependent dioxygenases) *AtGA2OX1* were selected to evaluate the changes in transcription. Under normal growth conditions, there was no significant difference in gene expression between WT plants and transgenic lines. Under Al stress, the expressions of *AtALMT1, AtALS3* and *AtGA2OX1* were upregulated in *GsGIS3* transgenic lines with 10-fold, 1.8-fold, and 3-fold improvement as compared to those in the WT plants, respectively (Figure 7A,B,E). The expressions of *AtGA3OX1* and *AtGID1* were downregulated to 0.6-fold in *GsGIS3* transgenic lines as compared to those in WT plants (Figure 7C,D).

## 3. Discussion

Plant growth and development are mainly constrained by Al rhizotoxicities on acid soil [3]. To adapt to these stress factors, multiple regulatory mechanisms involving a quantity of genes were discovered. The C2H2 ZFP family is one of the biggest transcription factor (TF) families in plants. Since Takatsuji discovered the first plant C2H2 zinc-finger protein gene *EPF1* [24], many C2H2 ZFP family genes have been proved to play various roles in the resistance to abiotic stresses and plant growth as shown by previous studies [23,29,41]. However, few studies have attempted to interpret the C2H2 ZFP members’ functional roles and underlying transcriptional regulation mechanism in soybean. In our study, we cloned a typical zinc-finger protein, *GsGIS3*, from *Glycine soja*. The amino acid sequences alignment showed that GsGIS3 protein contained a QALGGH motif and one typical zinc finger structure, which was most similar to three other C2H2 ZFPs AtGIS3 (NP_177003.1), AtZFP6 (NP_176873.1), GmZF1 (AAZ03389.1) (Figure 1A). The GsGIS3 protein was localized in the nucleus (Figure 2), which was identical with that of STOP1 in *Glycine max* [37] and *Vigna umbellate* [42]. Similar results were found in GIS3 and ZFP6 protein in *Arabidopsis* [43], ZFN1 in *Zoysia japonica* [34], and ZFP1 in *Glycine soja* [44]. As extensively reported, we speculated that GsGIS3 protein might function as a typical C2H2 ZFP and play certain roles in plants.

The evolutionary tree analysis of GsGIS3 protein and other C2H2 ZFPs in different species showed that GsGIS3 protein had more similarity to AtGIS3 and AtZFP6 in *Arabidopsis* (Figure 1B). The GsGIS3 protein belongs to the GIS subfamily of C2H2 ZFPs. Recent research confirmed that the members of GIS subfamily such as AtGIS3 and AtZFP6 play significant roles in stem and root development through GA and cytokinin signaling in *Arabidopsis* [43,45]. The *GsGIS3* gene, which is constitutive expression with the rich transcripts in soybean stem and root, was upregulated rapidly by AlCl_3_ and GA3 treatment (Figure 3). Therefore, we speculated that *GsGIS3* may be involved in the regulation of plant tolerance to Al stress through Gibberellins (GAs) pathways. To investigate its function, *GsGIS3* was transformed into *Arabidopsis* to obtain the homologous lines to verify the *Arabidopsis* tolerance to Al stress. The seedling root elongation of *GsGIS3* transgenic lines was found to exhibit better growth under Al stress than that of WT plants (Figure 4A and Figure 5A), which appeared in the form of higher total root length and total root surface area (Figure 5B,C). The phenotypes in response to Al stress were also consistent with other C2H2 ZFPs in different species. For instance, *GhSTOP1* could improve Al and H^+^ stress tolerance in transgenic cotton plants, and it played a role as an essential gene to regulate the expression of several genes that are necessary for the tolerance mechanisms in acid aluminum soil and lateral root development [46]. In addition, STOP1-like proteins were indicated to associate with Al tolerance in sweet sorghum. A previous report indicated that proline content in plants increases under stress, and a large amount of proline can alleviate the damage caused by adversity stress [20]. Proline and MDA can be used as physiological indicators for evaluating the resistance of plants to abiotic stress [21]. MDA is an important parameter to reflect the potential antioxidant capacity of plants, the content of MDA is significantly accumulated when the plants are under stresses. In this study, the contents of proline and MDA in *GsGIS3* transgenic lines and WT were also verified to monitor the resistance to Al stress, and the results were associated with phenotype observations. Transgenic lines showed higher proline but lower MDA contents than those in WT plants in Al stress (Figure 4B,C). This is similar to the previous research [34,47], indicating that over-expression of *GsGIS3* may improve the resistance to Al stress through increasing the content of proline and decreasing the content of MDA.

To further verify the function of *GsGIS3* in Al tolerance, homologous expression of *GsGIS3* in soybean hairy roots was done using Hematoxylin staining. In the previous research, hematoxylin could easily combine with Al in the root to form a purplish red complex. The degree of Al staining combined with hematoxylin and root tip was deep, which means that more Al was accumulated in the root tip, which was sensitive to Al toxicity. Due to the blue-purple color of hematoxylin when complexing with Al, the visual evaluation of dyed roots can be used to detect the accumulation of Al in root tissues [48]. In recent years, there have been more and more reports on the use of soybean hairy root transformation to verify the function of the soybean Al resistance gene. For example, the dyed *GmME1-OX* hairy root tip was shallow, and the content of Al was low, indicating strong Al resistance [49]. After Al treatment, the hairy roots of over-expression were found to have the shallow color compared to the control, with the deep color observed of RNAi roots compared to the control, which showed that the Al ion content of *GsGIS3-OX* hairy root was lower than control (Figure 6). This supported the assumption that over-expression of *GsGIS3* could enhance the resistance to Al stress.

Some studies have demonstrated that transcription factor could enhance tolerance to Al stresses through regulating the expression of Al-tolerance-related genes [37,42,50]. Several studies have reported that *ALMT1* gene is related to Al tolerance in *Arabidopsis* [18], *Brassica napus* [51], and rye [52]. To explore the potential molecular regulatory mechanisms of *GsGIS3* in Al stress responses, some Al-tolerance-related genes were assayed in wild-type and transgenic *Arabidopsis* under normal and stress conditions. The increased transcript level of *ALMT1* and *ALS3* showed an improved Al tolerance capability in *GsGIS3*-over-expressing lines (Figure 7A,B), which was consistent with *VuSTOP1* [42], *GmSTOP1* [37] and *WRKY46* [50]. *ALMT1*, which promotes malate secretion, is related to the most Al-tolerant phenotype in *Arabidopsis*, and previous reports indicated that transcription factor *WRKY46* could bind the promoter of *ALMT1* to modulate Al stress tolerance [50], suggesting that *GsGIS3* may function as a regulator of *ALMT1* to improve Al tolerance.

Recent studies on GAs have found that related genes and regulatory proteins in GAs synthesis and signaling pathways are closely related to the plant-stress-resistance. Plants can adapt to the adverse environment by regulating the expression of gibberellin-related metabolic genes to reduce the GA bioactivity in the plant, such as Gibberellin synthetase (GA20OXs and GA3OXs), Gibberellin inactivation enzyme (GA2OXs) and DELLAs [53,54,55]. A previous study has reported that NO induced the synthesis of auxin and gibberellin to promote the degradation of DELLA proteins, which improved the plant’s tolerance to aluminum toxicity [19]. Previous studies revealed that plants could improve their resistance to stress through reducing the accumulation of active gibberellin in plants [56], usually accompanied by the upregulation of *GA2OX1* and the downregulation of *GA3OX1* [57]. Similar results were also found in rice that over-expressed *GA2OX5* increased the salt tolerance of rice, and its salt tolerance disappeared when GA3 was applied. Studies in *Arabidopsis* found that in DDF1-overexpressing transgenic *Arabidopsis* lines, the transcription level of *GA2OX7* gene increased the resistance to salt [58]. In the study of the mitochondrial phosphate transporter (MPT) of *Arabidopsis*, the over-expressed MPT of *Arabidopsis* was found to be more sensitive to salt [59]. The research showed that the expression of *GA3OX1* increased in the transgenic plant, while the expressions of GA2OXs decreased [59]. Furthermore, GID1 is the receptor of the active Gibberellin, and the reduction of GID1 can also result in the decrease of active Gibberellin [60]. The abovementioned research indicated that the plant could enhance resistance to abiotic stress through GAs pathways. In our study, the transcription level of *GA2OX1* was upregulated by *GsGIS3*, whereas the expression of *GA3OX1* was reduced in transgenic *Arabidopsis* in Al stress (Figure 7C,D) and the expression of *AtGID1* was reduced in *GsGIS3* transgenic plants, which suggest that *GsGIS3* might promote plant tolerance to Al stress to a certain extent through GAs pathways.

## 4. Materials and Methods

### 4.1. Plant Materials and Growth Conditions

A type of *Glycine soja*, “BW69”, was used to clone *GsGIS3* for investigation of *GsGIS3* expression patterns in response to various stresses including AlCl_3_, NaCl, GA3, 6-BA and ABA. All wild soybean BW69 seeds were grown in an artificial climate chamber (22–24 °C temperature, 66% humidity with a 16-h/8-h light/dark photoperiod), as described in detail previously [61]. After germination, the seedings were pre-cultured in surface-sterilized vermiculite for 2 days and then transplanted into the 0.5 mM CaCl_2_ solution (pH 5.8) when the cotyledons unfolded. After adapting for two days, for the experiments on response to different stresses, the wild soybean seedlings were treated in the 0.5 mM CaCl_2_ solutions with 50 μM AlCl_3_ (pH 4.5), 200 mM NaCl, 100 μM GA3, 100 μM 6-BA or 10 μM ABA for 24 h (*n* = 16 seedings per group). The root samples (6 cm long) were obtained from the seedings treated after 0, 2, 4, 6, 8, 12 and 24 h [43,47,61]. To analyze the tissue expression pattern of *GsGIS3*, the seedings with open cotyledons were cultured in the nutrient solutions (pH 5.8, 0.5 mM CaCl_2_) for 4 days after treatment with 50 μM AlCl_3_ for 6 h. Samples of root (6 cm long), stem leaf and different root sections (0–2, 2–4 and 4–6 cm) were collected from the seedings (*n* = 16 seedings per group). To analyze the influence of the Al concentration gradient on gene expression, the seedings with open cotyledons were cultured in the nutrient solutions (pH 5.8, 0.5 mM CaCl_2_) for 4 days, and then the seedings were treated with 0, 25, 50, 75 and 100 μM AlCl_3_ (pH 4.5,0.5 mM CaCl_2_). Root samples (6 cm long) were harvested after 6 h treatments (*n* = 16 seedings per group). All samples were frozen with liquid nitrogen and stored at −80 °C [62]. *Arabidopsis* (ecotype Col-0) was used for *GsGIS3* genetic transformation and grown in the 1:1 (v/v) mixture of peat soil: vermiculite.

### 4.2. GsGIS3 Gene Isolation

Based on the previous analysis of acidic aluminum tolerance-related gene expression profiles (unpublished data), we obtained the sequence information of the *GsGIS3* gene from the database of the National Center for Biotechnology Information (NCBI) (https://www.ncbi.nlm.nih.gov/) with the protein ID of XP_003536676.1. The *GsGIS3* gene-specific primers were designed in accordance with the full-length cDNA, and the primers were used to isolate the gene from roots of BW69 by RT-PCR.

### 4.3. Bioinformatics Analysis

The homologous genes and proteins of GsGIS3 were searched in the database of NCBI and Joint Genome Institute (JGI) (https://phytozome.jgi.doe.gov/) according to the nucleotide sequence and the amino acid sequence of GsGIS3 protein. We aligned the nucleotide sequences with the software of DNAMAN. The prediction of the GsGIS3 protein conserved domain and structure was made using the NCBI blast results. We used the software of MEGA 7.0 to construct the phylogenetic trees by the Neighbor-Joining method; the number on the node represents the percentage of the boot value of 1000 replicates [63]. Its molecular weight (MW) and theoretical isoelectric point (pI) were computed through the Compute pI/MW tool (https://web.expasy.org/compute_pi/) [34]. We used an online website to speculate the localization of the protein (http://www.csbio.sjtu.edu.cn/bioinf/Cell-PLoc-2/).

### 4.4. Vector Construction and Transformation of GsGIS3 in Arabidopsis

We used the specific primers to amplify the full coding sequence of *GsGIS3* from the GsGIS3-pMD-18T vector, and then inserted it into the *Xba*I and *EcoR*I sites of pCAMBIA1301 by the restriction endonuclease. After the reforming reaction, the pCAMBIA1301-GsGIS3 fusion construct was produced under the action of the cauliflower mosaic virus 35S (CaMV 35S) promoter.

The complete coding sequence of GsGIS3 without a stop codon was inserted into the *Nco*I and *Spe*I restriction sites. Then, along with the green fusion protein (eGFP), a fusion construct of pCAMBIA1302-GsGIS3-eGFP was produced under the control of CaMV 35S promoter.

The sequence outside the conserved domain of GsGIS3 (206 bp) was inserted into the both ends of the RNAi vector pMU103 intron. The forward fragment was inserted into *Xma*I and *Sac*I sites first, and then the reverse fragment was inserted into the *Asc*I and *Avr*II sites.

### 4.5. Subcellular Localization of the GsGIS3 Protein

In accordance with previous research [40], localization of the GsGIS3 protein was performed. We transformed the plasmids into the *Agrobacterium tumefaciens* strain GV3101 by the heat shock method. Then, the *Agrobacterium tumefaciens* with plasmids was injected into the young leaves of 4-week-old tobacco plants in accordance with a previously described method [64]. After 48 h, we observed and photographed the lower epidermal cells of the leaves with a confocal laser scanning microscope (Leica, Germany).

### 4.6. RNA Extraction and Quantitative Real-Time PCR

The samples, including the wild soybean and *Arabidopsis*, were used with the TRIzol reagent (Invitrogen) method to extract the total RNA. Reversing transcription for single-stranded cDNA was synthesized with 1 mg total RNA using a HiScript III 1st Strand cDNA Synthesis Kit (Vazyme, NanJing, China). The qRT-PCR analyses were performed on a CFX96TM Real-Time system (United States) with ChamQ SYBR qPCR Master Mix (Vazyme, NanJing, China) in a total volume of 20 μL. A two-step qRT-PCR was adopted and set as follows: initial denaturation at 95 °C for 30 s, followed by 39 cycles of 95 °C for 10 s and 60 °C for 30 s. The gene *Actin3* or *tubulin* was chosen as the inner reference gene. Al resistance genes *ALMT1* (*At1G08430*) and *ALS3* (At1G08430) were selected to investigate transcriptional mechanisms of Al stress response. Genes related to Gibberellin anabolism and signal transduction of *GA3OX1* (*AT1G15550*), *GID1* (*AT3G05120*) and *GA2OX1* (*AT1G78440*) were chosen to surmise the potential regulatory mechanism. *β-tubulin* gene was adopted as internal reference. The analyses for qPCR results were assessed by the 2^–ΔΔCt^ method [65]. The specific primers are listed in Appendix A.

### 4.7. Generation of Transgenic Plants

Using the previously described floral dip method [66], *Agrobacterium tumefaciens* GV3101 transformed with construct plasmids was used to infect *Arabidopsis* plants to generate transgenic plants expressing GsGIS3. Transgenic *Arabidopsis* seeds were screened using, 20 mg/mL hygromycin. Positive transgenic plants were verified by reverse transcription PCR and genomic PCR. The T_3_ transgenic lines exhibiting 100% resistance to hygromycin were harvested for further phenotype research.

### 4.8. Acidic Aluminum Treatment in Transgenic Arabidopsis

For vegetative growth, all *Arabidopsis* seeds were surface-sterilized with 10% sodium hypochlorite for 10 min and then wash with sterilized ultra-pure water 5 times on a clean bench. The T_3_ generation *Arabidopsis* seeds were planted on 1/2 Murashige and Skoog (MS) (pH 5.8) agar medium. After the vernalization at 4 °C in darkness for 3 days, the seedings were transferred to the incubator (16 h light/8 h dark) at 22 °C for further treatment. For long-term AlCl_3_ treatment in 1/2 MS phytagel medium, the *Arabidopsis* seedlings with the consistent growing situation (1 cm long) in the 1/2 MS were selected to transfer to the medium containing 0, 50 and 100 μM AlCl_3_ (pH 4.5, 1/2MS) from the 1/2 MS phytagel medium (pH 5.8), and then the seedings were cultivated at 22 °C (16 h/8 h light/dark) for phenotype observation. For long-term AlCl_3_ treatment in nutrient solution, the *Arabidopsis* seedlings were cultivated in an artificial climate chamber (22–24 °C temperature, 66% humidity with a 16-h/8-h light/dark photoperiod), and, when root length was up to 0.8–1 cm, they were transferred to 1/30 Hoagland nutrient solution (without NH_4_H_2_PO_4_ and plus 1 mM CaCl_2_ and 2 μM AlCl_3_) for 21 days [42]. Main root elongation was measured in accordance with previously described method [67]. The whole seedlings were used to measure the root indicators (*n* = 20 seedings per group). The root measurements that included total root length and total root surface areas (SA) were done via WinRHIZO system (Regent Instruments, Québec, Canada).

### 4.9. Measurements for Proline and MDA

A previously described protocol was followed to measure the content of free proline in plants [68]. Briefly, 5 mL of 3% sulfosalicylic acid were added in 100 mg of plant samples when they were ground sufficiently. After cooling at room temperature, we transferred the 2 mL supernatant to a new test tube, adding 2 mL of acid ninhydrin and 2 mL of glacial acetic and mixing thoroughly. The mixture was heated in a boiling water bath for 40 min, and 5 mL of toluene were added when it was cooling. After delamination, the toluene was absorbed to measure the proline content at 520-nm absorbance. For MDA measurement, we also ground 100 mg plant samples with 5 mL of 5% trichloroacetic acid (TCA) in order to remove the 2 mL supernatant; 2 mL of 0.67% thiobarbituric acid (TBA) were added and boiled in the boiling water for 30 min after mixing well; and the MDA contents were measured at 450, 532 and 600 nm absorbance in accordance with the method described in detail previously [69].

### 4.10. Hematoxylin Staining in Soybean Hairy Roots

The pCAMBIA1301-GsGIS3 vector, RNAi vector and the empty pCAMBIA1301 vector were injected with *Agrobacterium rhizogenes* pathogenic strain K599 by electroporation [70], and the cotyledon Stage of WaYaoHuangDou (an aluminum sensitive type Soybean) plants were transformed using a syringe to inject the cotyledon. After several days of culturing under suitable temperature and humid environment, the hairy roots were harvested to study the Al ion accumulation degree of the soybean hairy roots by staining Hematoxylin [71]. After treating in the solution with AlCl_3_ (pH 4.5, 0.5 mM CaCl_2_) for 6 h, the roots were transferred into ultra-pure water to clean for 30 min, followed by staining with hematoxylin for 30 min. Lastly, the roots were cleaned again in ultra-pure water for 30 min, the hairy roots phenotypes were observed by the microscope (Leica, Germany).

### 4.11. Statistical Analysis

All data were analyzed by the mean ± SD of three biological replicates using SPSS version 20.0 (IBM, Chicago, IL, United States). Student’s test at *P* = 0.01 or *P* = 0.05 was used to analyze the significant difference between observation values.

## 5. Conclusions

We isolated a wild soybean C2H2 zinc-finger protein gene, *GsGIS3*, which is upregulated in response to AlCl_3_, and GA3 and is enriched in the stem and root of soybean. The GsGIS3 protein is located in the nucleus. Heterologous overexpression of *GsGIS3* in *Arabidopsis* enhances the resistance of transgenic plants to Al stress with MDA content reducing and proline accumulation. The analysis of molecular mechanisms showed that the enhanced resistance to Al stress in *Arabidopsis* might result in the comprehensive roles in relation to transcriptional activities of the Al tolerance related genes and/or GAs related genes. Therefore, the results above indicate that *GsGIS3* may enhance the resistance to Al stresses through certain pathways related to Al stress and/or GAs.

## Figures and Tables

**Figure 1 ijms-21-02754-f001:**
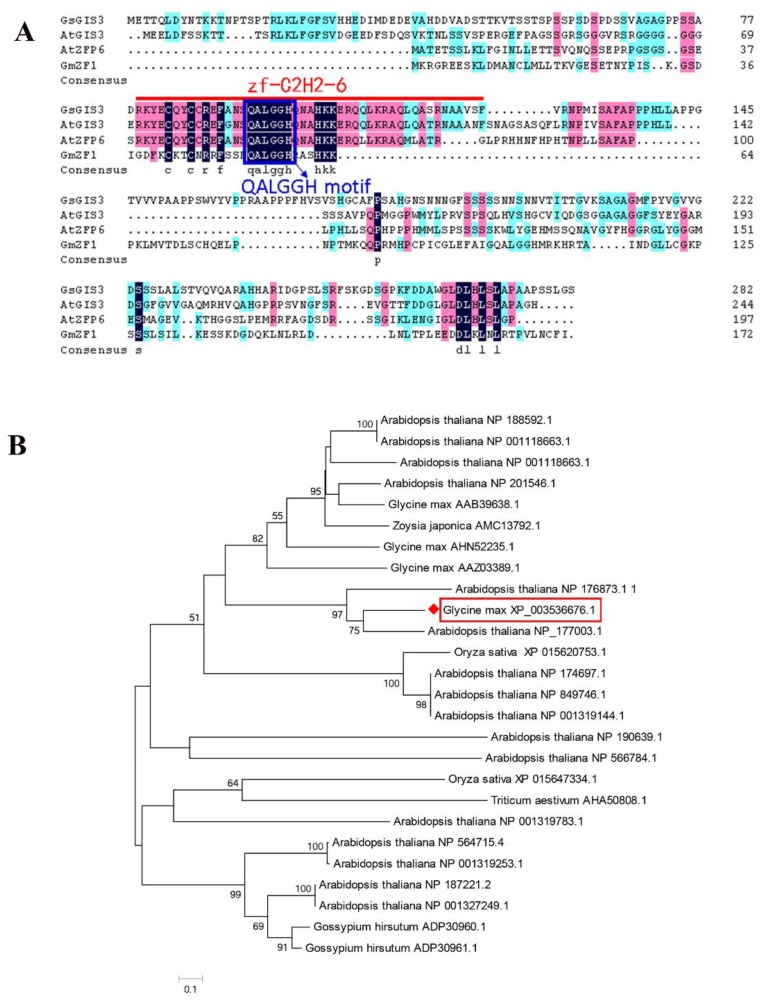
Amino acid sequence and phylogenetic analysis of GsGIS3. (**A**) The alignment of C2H2 zinc finger proteins. The position of the domain of zinc finger is labeled. GsGIS3 (XP_003536676.1), AtGIS3 (NP_177003.1), AtZFP6 (NP_176873.1), and GmZF1 (AAZ03389.1). (**B**) Phylogenetic analysis of GsGIS3 protein orthologs. The GsGIS3 protein is marked with a red box.

**Figure 2 ijms-21-02754-f002:**
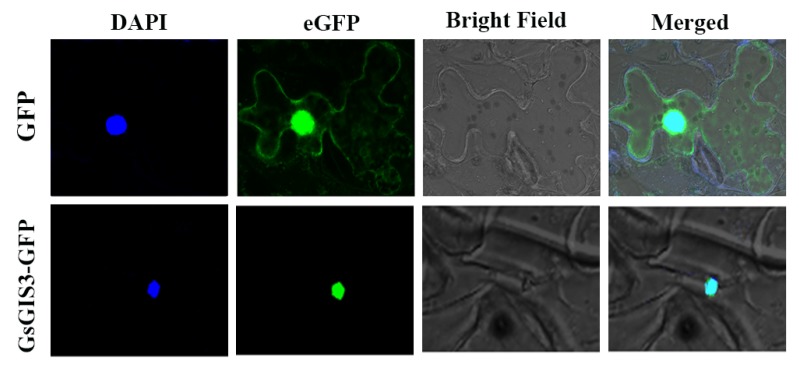
Subcellular localization of GsGIS3 protein. Subcellular localization of GsGIS3 protein in leaf epidermal cells of tobacco. The protein expression of GFP or GsGIS3-GFP from the leaves after agro-infiltration for 48 h was visualized using a confocal laser scanning microscope (Leica, Germany) [39].

**Figure 3 ijms-21-02754-f003:**
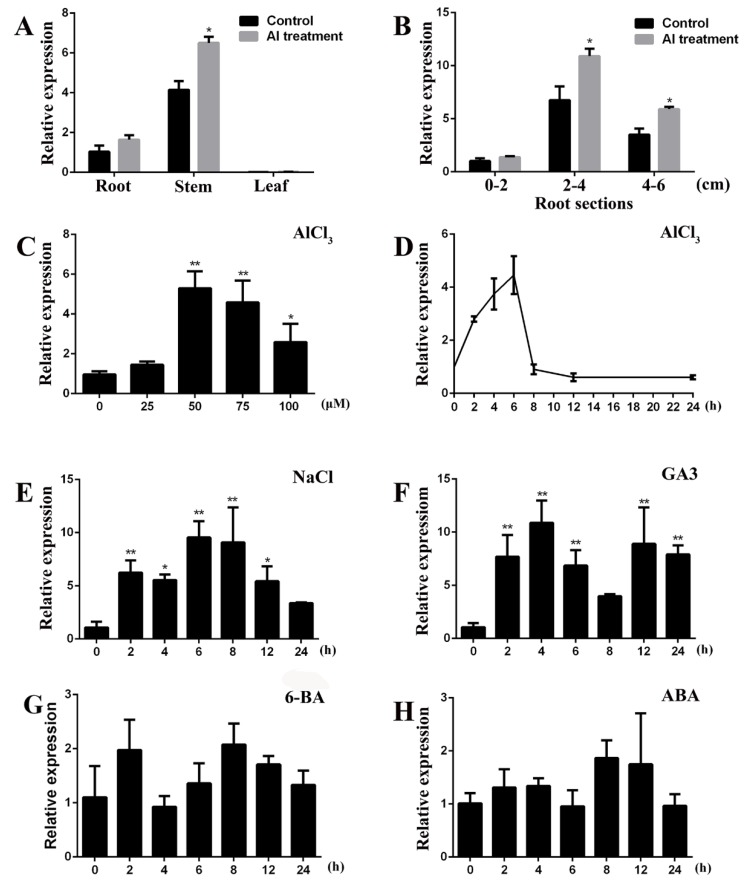
Expression characteristics of *GsGIS3.* (**A**) *GsGIS3* expression levels in root, stem, and leaf of *Glycine soja.* (**B**) Expression levels of *GsGIS3* in *Glycine soja* roots in different sections. (**C**) Pattern of *GsGIS3* expression under Al concentration gradients. (**D**) Temporal expression pattern of *GsGIS3* under acidic aluminum exposure. (**E**–**H**) Patterns of *GsGIS3* expression under the different conditions of abiotic stresses: 200 mM NaCl (**E**); 100 μM GA3 (**F**); 100 μM 6BA (**G**); and 10 μM abscisic acid (ABA) (**H**). *GsGIS3* transcript abundance was assessed by qRT-PCR using the 2^–ΔΔCt^ method with the actin *Actin3* gene as an internal control [40]. The data are represented as the averages of three independent biological experiments ± SD, and asterisks indicate a significant difference (* *P <* 0.05; * *P <* 0.01) compared with the corresponding controls.

**Figure 4 ijms-21-02754-f004:**
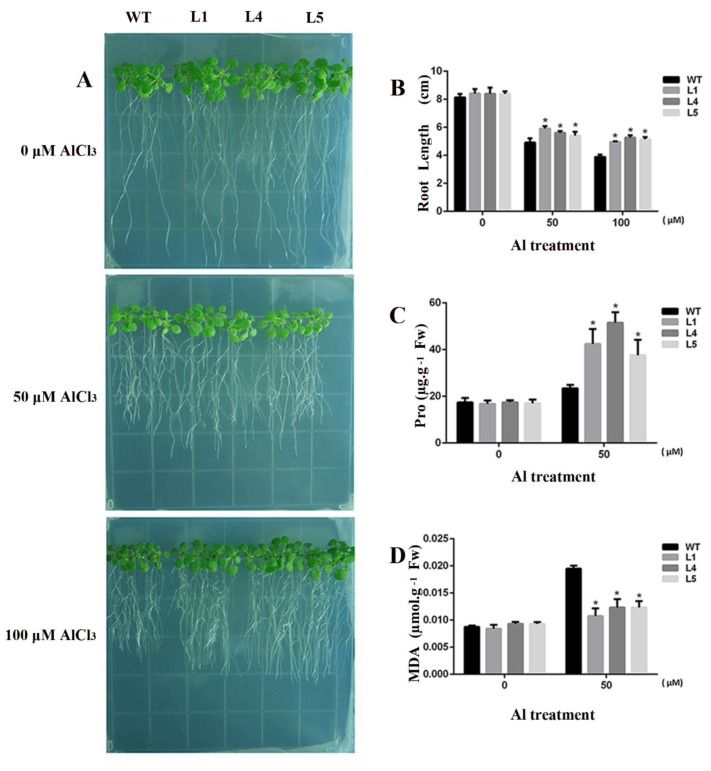
Overexpression of *GsGIS3* in *Arabidopsis* enhanced Al tolerance. (**A**) The phenotypes of transgenic lines and control under AlCl_3_ treatment for seven days. (**B**) Statistical analysis of root elongation. (**C**) The determination of proline content. (**D**) The determination of MDA content. The vertical columns for the average observation value of the three repetitions represent the means ± SD. Three independent biological experiments were carried out to investigate the status of seedlings and accumulations of proline and MDA in plants of WT and *GsGIS3* transgenic lines under Al stress. Asterisks indicate significant differences between WT and *GsGIS3* transgenic lines (* *P* < 0.05). WT, wild-type of *Arabidopsis* (Col-0); L1, L4 andL5, *GsGIS3 Arabidopsis* transgenic lines of T_3_ generations.

**Figure 5 ijms-21-02754-f005:**
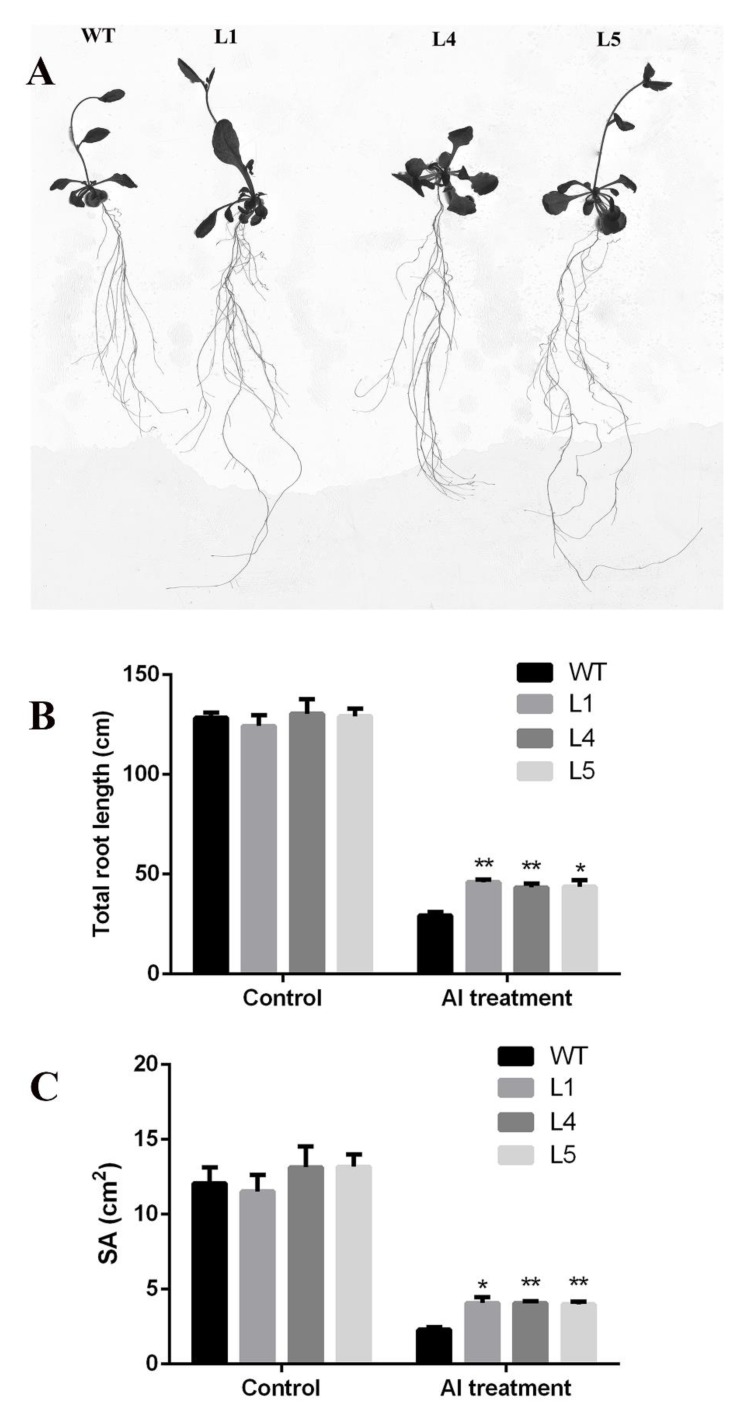
Phenotype observation and indexes determination. (**A**) The phenotype of transgenic lines and control in 1/30 Hoagland nutrient solution with 2 μM AlCl_3_. (**B**) The total root length. (**C**) The total root surface area. Asterisks indicate significant differences between WT and *GsGIS3* transgenic lines (* *P* < 0.05; ** *P* < 0.01). WT, wild-type of *Arabidopsis* (Col-0); L1, L4 and L5 *GsGIS3 Arabidopsis* transgenic lines of T_3_ generations; SA, the total root surface area. Three independent biological experiments were carried out to investigate the status of seedlings and measure the root indexes.

**Figure 6 ijms-21-02754-f006:**
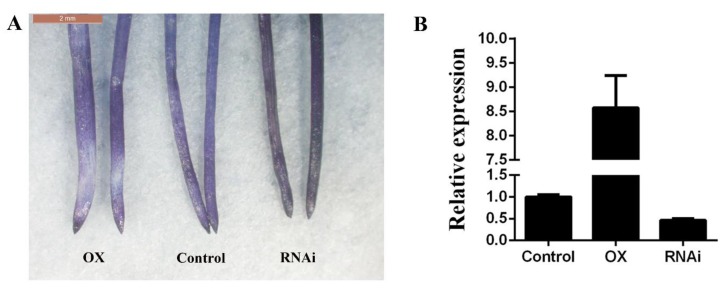
Hematoxylin staining in soybean hairy roots. (**A**) Hematoxylin staining in soybean hairy roots. (**B**) The RNA molecular level identification. OX, *GsGIS3*-overexpressing transgenic soybean hairy roots; RNAi, *GsGIS3*-RNAi in soybean hairy roots; Control, Agrobacterium rhizogenes pathogenic strain K599 in soybean hairy roots. Three independent biological experiments were carried out to investigate the status of seedlings and measure the root data.

**Figure 7 ijms-21-02754-f007:**
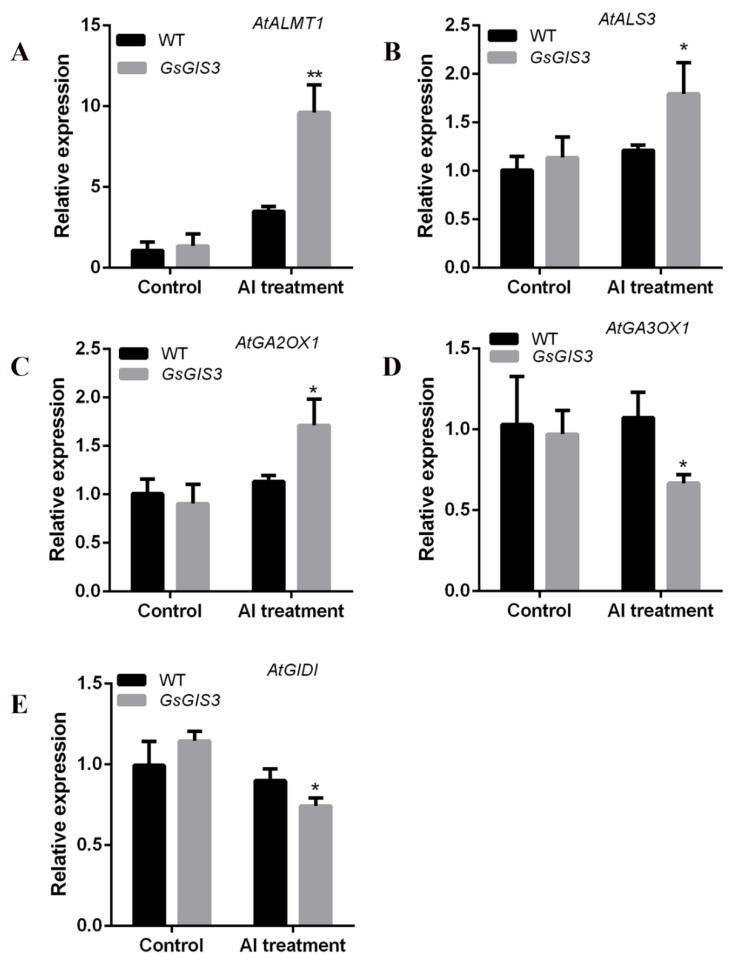
Expression Patterns of Al Stress/GA3-related Genes Regulated by *GsGIS3*: (**A**) *AtALMT1* expression; (**B**) *AtALS3* expression; (**C**) *AtGA2XO1* expression; (**D**) *AtGA3OX1* expression; and (**E**) *AtGID1* expression. Asterisks indicate a significant difference (* *P <* 0.05; ** *P <* 0.01) compared with the controls.

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
