# Peer review of "Heterologous Expression of a Glycine soja C2H2 Zinc Finger Gene Improves Aluminum Tolerance in Arabidopsis"

_ijms, 2020, doi:10.3390/ijms21082754_

Round 1

Reviewer 1 Report

Heterologous Expression of a Novel Glycine soja C2H2 Zinc Finger Gene, GsGIS3, Improves Aluminum Tolerance in Arabidopsis

The authors have attempted to functionally characterise a zinc finger transcription factor from Glycine soja – a wild relative of soybean which is Al tolerant. They have cloned the gene; carried out phylogenetic analysis; subcellular localisation and other plant experiments. The whole manuscript is plagued with poor sentence structure and grammar; not enough experimental detail is provided and conclusions are overstated.

The authors, claim the GsGIS3 is involved in Al tolerance but based on the data presented, it would suggest that this gene is more likely to be involved in abiotic stress tolerance than Al tolerance alone. For e.g. the authors, have quantified the expression of the gene in response to Al, NaCl, GA3 and other compounds but justification for such experiments is limited apart from the fact that C2H2 TF may be involved in such stresses. Further in spite of the experimental data showing increased activity of GsGIS3 in response to both Al and NaCl, authors conclude that this gene improves Al tolerance. It would be better to perform experiments with different concentrations of NaCl to investigate the role of gene in salinity.

In the section on bioinformatics analysis (2.1), No accession numbers provided for Fig 1A; in Figure 1B-it is hard to see where the GsGIS3 is located or even if it has been included in the figure? Names of the genes should be included in brackets to make it easier for the readers. Legends need to improved, by including more details.

In the section on subcellular localisation (2.2): DAPI has been used to stain for nuclear localisation, however it is a strange that DAPI staining of GsGIS3-GFP is barely visible compared to the controls! This figure needs to be redone with better images.

Section 2.3 looks at expression patterns of GsGIS3 via Q PCR. The results need to be explained more clearly. To start with concentrations of treatments need to be clearly stated for Al treatment. Figure 3B- expression of the gene seems to increase 2-4 cm from the root apex in both controls and treatment; authors need to clearly state why this is so? In Figure 3C, where was the expression of the gene measured in the root-was it in the total roots or sections? It needs to be made clearer in the text as to why authors felt the need to test compounds such as NaCl, GA and ABA.

In section 2.4, what was the age of seedlings subjected to Al treatments?  How many seedlings per replicate per treatment were tested in the experiments? Figure 4B and C- it is very difficult to see any difference in total root lengths; all seem inhibited by Al treatments including controls. It is unclear how root elongation was calculated, is it more a measurement of total root lengths? Further it not stated why the authors chose to use only 2 uM AlCl3 in this experiment and also omitted NH4H2PO4. What are the reasons for dropping the concentration of AlCl3? In the text authors state that MDA concentrations in WT plants decreased (page 7, L169-170), but the figure actually shows an increase? Further how are the stats done in the figures-is the comparison across treatments or within treatments? All details need to be included.

No justification in the text for choosing the three transgenic lines for evaluation is provided. I struggle to see the relevance of the results presented in Figure 5. Al treatment inhibits growth in both controls and transgenic plants. How many seedlings per replicate per treatment were tested in the experiments?

In figure 6, hematoxylin staining of roots has been carried out to demonstrate Al accumulation. However authors failed to mention where the RNAi roots come from, how was that generated. The results from this experiment are subjective, hard to be convinced that OX improves Al tolerance.

Overall the manuscript needs to rewritten to address poor sentence structure and grammar; more details need to be provided in the text and figure legends to improve the quality of the manuscript. The discussion section needs to be improved and the authors need to re-evaluate their conclusions regarding the role of GsGIS3 in Al tolerance.

The following (highlighted in red) are only a few examples of poor English; too many to list them all:

15 In this study, a ZFP

16 transcription gene, GsGIS3, cloned from wild soybean was functionally verified to Al tolerance in

17 Arabidopsis.

31 When the soil with a pH below 5, Al mainly exists as Al3+ which forms Al complex in the soil ionic

32 [3,4], The Al3+ can inhibit root development by damaging cell walls [5] and cytoskeletons [6],

40 exclude Al3+ outside from the protoplasts by some ways, while internal tolerance mechanism is to – not sure what the authors mean here?

76 was induced by drought and cold and the over-expressed transgenic plants showed resistance to low

77 temperature and low temperature [33]. Similar to rice and Arabidopsis, many ZFP genes which were

78 certified specifically in involved to abiotic stress have been reported in other species. For instance, not sure what the authors mean here?

94 little information is involved in Al tolerance in soybean. Not true as authors mention studies in soybean in lines 87-89

119 Laser confocal microscopic observations showed strong green fluorescence signal in the whole cells

120 of the GFP while the green fluorescence signal was only found in nucleus in pCAMBIA1302-GsGIS3-

121 GFP cells (Figure 2). The results certificated

Author Response

Dear editors and reviewers,

We earnestly appreciate time and effort taken by the Editors and Reviewers to improve the content and clarity of our manuscript (ID ijms-699410). We have studied the comments carefully and have made corrections which we hope meet with your approval. According to your suggestion, we would have our manuscript undergone extensive English editing by the MDPI English editing service and submitted track-changed versions of manuscript. The main corrections in the manuscript and the response to each of the reviewer’s comments are detailed below. Thank you very much for your thoughtful comments and suggestions. We look forward to hearing from you regarding the suitability of the manuscript for publication in International Journal of Molecular Sciences.

Best regards,

Xiuxiang Zhang, PhD

Email: xiuxiangzh@scau.edu.cn

The authors have attempted to functionally characterise a zinc finger transcription factor from Glycine soja – a wild relative of soybean which is Al tolerant. They have cloned the gene; carried out phylogenetic analysis; subcellular localisation and other plant experiments. The whole manuscript is plagued with poor sentence structure and grammar; not enough experimental detail is provided and conclusions are overstated.

The authors, claim the GsGIS3 is involved in Al tolerance but based on the data presented, it would suggest that this gene is more likely to be involved in abiotic stress tolerance than Al tolerance alone. For e.g. the authors, have quantified the expression of the gene in response to Al, NaCl, GA3 and other compounds but justification for such experiments is limited apart from the fact that C2H2 TF may be involved in such stresses. Further in spite of the experimental data showing increased activity of GsGIS3 in response to both Al and NaCl, authors conclude that this gene improves Al tolerance. It would be better to perform experiments with different concentrations of NaCl to investigate the role of gene in salinity.

Response: Thanks for your suggestion. Our experimental data showed that GsGIS3 in response to both Al and NaCl. We also performed experiments with different concentrations of NaCl to investigate the role of gene in salinity. The methods were as follow:

“The surface-sterilized Arabidopsis seeds of wild type and transgenic lines were sown on the solid media of 1/2 MS. The 3-day-old seedlings from 1/2 MS medium were transferred to the plates of 1/2 MS containing NaCl (0, 50, 100 and 200 mM). The status of seedlings (n=15 seedings per group) was taken by photos and root lengths were measured when Arabidopsis plants were treated under NaCl stress for 7 days. Three independent biological experiments were carried out to investigate the status of seedlings and root growth of WT and GsGIS3 transgenic lines under NaCl stress.”

However, no difference was observed between wild type and transgenic lines in all three independent biological experiments. Therefore, we did not show the details of the experiments with different concentrations of NaCl in the manuscript.

Response to Reviewer Comments

General comments

Point 1: In the section on bioinformatics analysis (2.1), No accession numbers provided for Fig 1A; in Figure 1B-it is hard to see where the GsGIS3 is located or even if it has been included in the figure? Names of the genes should be included in brackets to make it easier for the readers. Legends need to improved, by including more details.

Response 1: Thanks for your suggestion. We added the accession numbers for Figure 1A in the annotation, the GsGIS3 protein was marked by a red box and it could be seen clearly. The changes were as follow:

Lines 120-122. “GsGIS3 (XP_003536676.1), AtGIS3 (NP_177003.1), AtZFP6 (NP_176873.1), GmZF1 (AAZ03389.1).”

“The GsGIS3 protein was marked with a red box.”

Point 2:  In the section on subcellular localisation (2.2): DAPI has been used to stain for nuclear localisation, however it is a strange that DAPI staining of GsGIS3-GFP is barely visible compared to the controls! This figure needs to be redone with better images.

Response 2: Thanks for your suggestion. Line 132. We have redone the figure in the manuscript, and now the nucleus spot of GsGIS3-GFP is visible.

Point 3: Section 2.3 looks at expression patterns of GsGIS3 via Q PCR. The results need to be explained more clearly. To start with concentrations of treatments need to be clearly stated for Al treatment.

Response 3: Thanks. As your suggestion. We added the statement for concentrations of treatments. Lines 150-152. “, and the expression level decreased to 2.5-fold at the treatment of 100μM AlCl3 compared with those detected under control condition.”

Point 4:  Figure 3B- expression of the gene seems to increase 2-4 cm from the root apex in both controls and treatment; authors need to clearly state why this is so?

Response 4: Thanks. In Figure 3B. The significant differences were compared between controls and treatment in each group (0-2 cm, 2-4 cm 4-6 cm). And we recalculated significant differences between controls and treatment. The expression of the gene had no change.

Point 5: In Figure 3C, where was the expression of the gene measured in the root-was it in the total roots or sections?

Response 5: Thanks. The expression of the gene measured in the total roots. Line 337. We added the description:

“the root samples (6 cm long) were obtained from…”

Point 6: It needs to be made clearer in the text as to why authors felt the need to test compounds such as NaCl, GA and ABA.

Response 6: Thanks. Many C2H2 ZFP family genes have been proved to play various roles in the resistance to abiotic stresses as shown by previous studies. According to several proteins with high homology with GsGIS3 protein such as AtGIS3, AtZFP6, we speculated the potential function of GsGIS3 and tried the corresponding test. Thanks for your suggestion. We added some sentences in the text to explain the reason for the test. Lines 150-152:

“To investigate the expression patterns of GsGIS3 response to other abiotic stresses, we analyzed the transcript abundance of GsGIS3 under NaCl, GA3, 6-BA and ABA treatments.”

Point 7: In section 2.4, what was the age of seedlings subjected to Al treatments?  How many seedlings per replicate per treatment were tested in the experiments?

Response 7: Thanks. About 3 days after the vernalization when the seedlings length to 1 cm, the seedings were subjected to Al treatment. Line 413. We added it in the sentence as follow:

“…with the consistent growing situation (1 cm long) in the…”

Point 8: Figure 4B and C- it is very difficult to see any difference in total root lengths; all seem inhibited by Al treatments including controls. It is unclear how root elongation was calculated, is it more a measurement of total root lengths?

Response 8: Thanks. The statistical data of the root is the root length not root elongation. We recalculated the root length. Under AlCl3 treatment, all seedings were inhibited. However, the transgenic lines showed better growth than wild-type. We have revised the relative sentences as follow:

Line 176 “…compared to the control, the root length of WT plants…”

Line 177 “…while the root lengths of GsGIS3 transgenic…”

Point 9: Further it not stated why the authors chose to use only 2 uM AlCl3 in this experiment and also omitted NH4H2PO4. What are the reasons for dropping the concentration of AlCl3?

Response 9: Thanks. To the long-term AlCl3 treatment in nutrient solution, Arabidopsis hardly survives in high-concentration aluminum solution. Before that we tried to cultivate the seedings in 20 μM AlCl3 solution, the roots of the transgenic lines and wild-type were both limited. “PO4-3” would be hydrolyzed with Al3+ in the solution. Therefore, we removed “NH4H2PO4”. The method of this experiment referred to the following references.

“Low pH, aluminum, and phosphorus coordinately regulate malate exudation through GmALMT1 to improve soybean adaptation to acid soils. Plant Physiology 2013, 161, 1347-1361.”

“Characterization of an inducible C2H2‐type zinc finger transcription factor Vu STOP 1 in rice bean (Vigna umbellata) reveals differential regulation between low pH and aluminum tolerance mechanisms. New Phytologist 2015, 208, 456-468.”

Point 10: In the text authors state that MDA concentrations in WT plants decreased (page 7, L169-170), but the figure actually shows an increase? Further how are the stats done in the figures-is the comparison across treatments or within treatments? All details need to be included.

Response 10: Thanks. We have revised the sentence to make it clear. Line183-186:

“In addition, under 100 mM AlCl3 treatment, MDA accumulation in WT plants improved to 0.019 μmol.g-1 of fresh weight while MDA contents in GsGIS3 transgenic lines increased to.the range from 0.010 mmol.g-1 to 0.013 mmol.g-1 of fresh weight in Al stress (Figure 4D). MDA in wild type was significantly higher than that of transgenic lines.”

Point 11: No justification in the text for choosing the three transgenic lines for evaluation is provided.

Response 11: Thanks. As your suggestion, we have added the reason for choosing the three transgenic lines for evaluation.

Lines 167-168: “we identified the transcription levels of five homozygous transgenic lines, and selected three lines with higher expression levels for subsequent experiments.”

Point 12: I struggle to see the relevance of the results presented in Figure 5. Al treatment inhibits growth in both controls and transgenic plants. How many seedlings per replicate per treatment were tested in the experiments?

Response 12: Thanks.The toxicity of aluminium to plants is mainly showed in the root, the root data reflects the toxicity of aluminium to plants. Both transgenic lines and wild-type Arabidopsis are inhibited when exposed to aluminium stress. The root measurements included total root length and total root surface areas (SA), were done via WinRHIZO system (Regent Instruments, Québec, Canada). The total length and surface of transgenic lines were better than wild-type, to a certain degree,it showed that the transgenic lines were more resistant to aluminium stress than wild -type.

20 seedings were tested in the experiments per group. As your suggestion. Lines 419-420. We added it in the manuscript:

“The whole seedlings were used to measure the root indicators (n=20 seedings per group).”

Point 13: In figure 6, hematoxylin staining of roots has been carried out to demonstrate Al accumulation. However authors failed to mention where the RNAi roots come from, how was that generated. The results from this experiment are subjective, hard to be convinced that OX improves Al tolerance.

Response 13: Thanks for your suggestion. Lines 386-388. In section 4.4. We added the step of construction of RNAi vector. The sentences were as follow:

“The sequence outside the conserved domain of GsGIS3 (206 bp) was inserted to the both ends of of RNAi vector pMU103 intron. The forward fragment was inserted into XmaI and SacI sites first, then, the reverse fragment was inserted into the AscI and AvrⅡ sites.”

Overall the manuscript needs to rewritten to address poor sentence structure and grammar; more details need to be provided in the text and figure legends to improve the quality of the manuscript. The discussion section needs to be improved and the authors need to re-evaluate their conclusions regarding the role of GsGIS3 in Al tolerance.

Specific comments

The following (highlighted in red) are only a few examples of poor English; too many to list them all:

15 In this study, a ZFP

16 transcription gene, GsGIS3, cloned from wild soybean was functionally verified to Al tolerance in

17 Arabidopsis.

Response: Thanks for your suggestion. Lines 15-18, we have rewritten the sentence and the sentence was as follow:

“In this study, GsGIS3 was isolated from Al-tolerant wild soybean gene expression profiles to be functionally characterized in Arabidopsis.”

31 When the soil with a pH below 5, Al mainly exists as Al3+ which forms Al complex in the soil ionic

Response: Thanks. Line 32. We deleted “ionic”.

32 [3,4], The Al3+ can inhibit root development by damaging cell walls [5] and cytoskeletons [6],

Response: Thanks. Line 33. We added “the structure of” before cytoskeletons.

40 exclude Al3+ outside from the protoplasts by some ways, while internal tolerance mechanism is to – not sure what the authors mean here?

Response: Thanks. Line 41. We deleted “outside”.

76 was induced by drought and cold and the over-expressed transgenic plants showed resistance to low

77 temperature and low temperature [33]. Similar to rice and Arabidopsis, many ZFP genes which were

Response: Thanks for your suggestion. We have corrected the incorrect description. Lines 78-79. The sentence was revised as follow:

was induced by drought and cold and the over-expressed transgenic plants showed resistance to low temperature and drought.”

78 certified specifically in involved to abiotic stress have been reported in other species. For instance, not sure what the authors mean here?

Response: Thanks for your suggestion. Line 80. We revised the sentence as follow:

“Similar to rice and Arabidopsis, many ZFP genes which were in involved in abiotic stress have been reported in other species.”

94 little information is involved in Al tolerance in soybean. Not true as authors mention studies in soybean in lines 87-89

Response: Thanks. We deleted this sentence to make it clear.

119 Laser confocal microscopic observations showed strong green fluorescence signal in the whole cells

120 of the GFP while the green fluorescence signal was only found in nucleus in pCAMBIA1302-GsGIS3-

121 GFP cells (Figure 2). The results certificated

Response: Thanks. We have revised the sentence as follow:

“By the laser confocal microscopic observations, pCAMBIA1302-GsGIS3-GFP protein accumulated mainly in the nucleus with strong signals of green fluorescence. In contrast, GFP alone was distributed evenly throughout all parts of the cell including the nucleus and cytoplasm (Figure 2). The results showed that…”

Reviewer 2 Report

The manuscript needs extensive revision and English language editing.

I recommend doing major revision for the following reasons:

  • Unidentified abbreviations as in the line: 49 “DELLA” and Line: 69 “SA”
  • Some sentences need to rewrite to be understandable as “A type of Glycine soja (wild soybean), “BW69” was used to cloned GsGIS3 for expression patterns analysis.” In section 4.1, line 318
  • section 4.1 needs to be rewritten as it is not understandable particularly stress treatment. Also, growth conditions lack some important details as the percentage of humidity and type of nutrients solution used, Hogland's or Ashton’s or what?
  • Section 4.3 is not described well. Steps of identifying GsGIS3 homologs, P-value cut-off, conserved domain and again the problem of language.
  • Some wrong typos are highlighted in the PDF file of the manuscripts.
  • Section 4.6 lacks important information about PCR steps as the volume of reaction and constituents as a template, primer conc., master mix, water and annealing temp and no. of cycles.
  • Did you do a standard curve for each gene? What is the standard curve series?
  • Scientific names in many locations throughout the manuscript are not written italics
  • In section 2.2, the authors declared that “Bioinformatics analysis predicted that GsGIS3 protein existed a localization signal in nuclear” however they did not mention any bioinformatics tools for the prediction of the subcellular localization in materials and methods.
  • Figure 2, images without scale bars
  • I am not sure about the significance in Figure 3G, please check it as the error bar at 8h is large and I can see that there is no significant difference between 0 and 8h.
  • (-1) in the title of the Y-axis in figure 4 should be superscript.
  • Total root length in figure 5B is not described in materials and methods and how it was calculated for all treatment?
  • More details about the mode of action of Hematoxylin staining. Why it is shallower or deeper?
  • I do not recommend using the word “novel” in the title.

Author Response

Dear editors and reviewers,

We earnestly appreciate time and effort taken by the Editors and Reviewers to improve the content and clarity of our manuscript (ID ijms-699410). We have studied the comments carefully and have made corrections which we hope meet with your approval. According to your suggestion, we would have our manuscript undergone extensive English editing by the MDPI English editing service and submitted track-changed versions of manuscript. The main corrections in the manuscript and the response to each of the reviewer’s comments are detailed below. Thank you very much for your thoughtful comments and suggestions. We look forward to hearing from you regarding the suitability of the manuscript for publication in International Journal of Molecular Sciences.

Best regards,

Xiuxiang Zhang, PhD

Email: xiuxiangzh@scau.edu.cn

Response to Reviewer Comments

Point 1: Unidentified abbreviations as in the line: 49 “DELLA” and Line: 69 “SA”

Response 1: Thanks for your suggestion. Line49 “DELLA”: Among the discovered plant gibberellin (GA) signaling molecules, there is a class of N-terminally highly conserved DELLA domains called DELLA family proteins which plays a role in suppressing GA signal transduction. Line 69 “SA”: the abbreviation of the Salicylic Acid. We added the explanations of abbreviations, the sentences were revised as follow:

“Previous study has reported that NO induced the synthesis of auxin and gibberellin to promote the degradation of the GA signal transduction inhibitory protein (DELLA proteins), which improved the plant's tolerance to aluminum toxicity [1].”

“Overexpression of ZAT18 enhanced drought tolerance of Arabidopsis [2], whereas AtZAT6 could regulate biotic and abiotic stress tolerance through the Salicylic Acid (SA) activation pathway [3].”

Point 2: Some sentences need to rewrite to be understandable as “A type of Glycine soja (wild soybean), “BW69” was used to cloned GsGIS3 for expression patterns analysis.” In section 4.1, line 318

section 4.1 needs to be rewritten as it is not understandable particularly stress treatment. Also, growth conditions lack some important details as the percentage of humidity and type of nutrients solution used, Hogland's or Ashton’s or what?

Response 2: Lines 317-318. Thanks. As your suggestion, we added the experimental details and the rewritten section was as follow:

“A type of Glycine soja, “BW69” was used to cloned GsGIS3 for investigation of GsGIS3 expression patterns in response to various stresses including AlCl3, NaCl, GA3, 6-BA and ABA. All BW69 seeds were grown in artificial climate chamber (22-24 °C temperature, 66% humidity with a 16-h/8-h light/dark photoperiod) as described in detail previously [4]. After germination, the seedings were pre-cultured in surface-sterilized vermiculite for 2 days and then transplanted into the 0.5 mM CaCl2 solution (pH 5.8) when the cotyledons unfold. After adapting for 2 days, for the experiments to response to different stresses, the wild soybean seedlings were treated in the 0.5 mM CaCl2 solutions with 50 μM AlCl3 (pH 4.5), 200 mM NaCl, 100 μM GA3, 100 μM 6-BA or 10 μM ABA for 24 h (n=16 seedings per group), the root tip samples (6 cm long) were obtained from the seedings treated after 0, 2, 4, 6, 8, 12 and 24 h were collected ,respectively [4-6]. To analyze the tissue expression pattern of GsGIS3, the seedings with open cotyledons were cultured in the nutrient solutions (pH 5.8,0.5 mM CaCl2) for 4 days, after treatment with 50 μM AlCl3 for 6 h, Samples of root (6 cm long) stem leaf and different root sections (0-2, 2-4, 4-6 cm) were collected from the seedings (n=16 seedings per group). To analyze the influence of the Al concentration gradient on gene expression, the seedings with open cotyledons were cultured in the nutrient solutions (pH 5.8,0.5 mM CaCl2) for 4 days, then the seedings were treated with 0, 25, 50, 75 and 100 μM AlCl3 (pH 4.5,0.5 mM CaCl2), root samples (6 cm long) were harvested after 6 h treatments (n=16 seedings per group). All samples were frozen with liquid nitrogen and stored at -80 °C [7]. Arabidopsis (ecotype Col-0) was used for GsGIS3 genetic transformation and grown in the 1:1 (v/v) mixture of peat soil: vermiculite.”

Point 3: Section 4.3 is not described well. Steps of identifying GsGIS3 homologs, P-value cut-off, conserved domain and again the problem of language.

Response 3: Thanks. As your suggestion, we revised some sentences. The sentences were as follow:

Line 359 “The homologous genes and proteins of GsGIS3 were searched from…”

Lines 362-363 “The prediction of the GsGIS3 protein conserved domain and structure were using the NCBI blast results.”

Point 4: Some wrong typos are highlighted in the PDF file of the manuscripts.

Response 4: Thanks. We revised the wrong typos one by one in the manuscripts.

Point 5: Section 4.6 lacks important information about PCR steps as the volume of reaction and constituents as a template, primer conc., master mix, water and annealing temp and no. of cycles.

Response 5: Thanks. Lines 388-390. We added the volume and program in the manuscripts, and the primers were listed in Supplementary Table S1. The sentence was as follow:

“…in a total volume of 20 μL. A two-step qRT-PCR was adopted and set as follows: initial denaturation at 95°C for 30 s, followed by 39cycles of 95°C for 10 s and 60°C for 30 s.”

Point 6: Did you do a standard curve for each gene? What is the standard curve series?

Response 6: Thanks for your suggestion. We did the standard curve for each gene. The standard curve series R2 were all around 0.998. Due to the length of article, we did not present the detail in the manuscript.

Point 7: Scientific names in many locations throughout the manuscript are not written italics

Response 7: Thanks. We revised the mistakes in the manuscripts.

Point 8: In section 2.2, the authors declared that “Bioinformatics analysis predicted that GsGIS3 protein existed a localization signal in nuclear” however they did not mention any bioinformatics tools for the prediction of the subcellular localization in materials and methods.

Response 8: Thanks. We added the methods in the manuscript and the sentences were revised as follow:

Lines 122-123. “We predicted that GsGIS3 protein was localized in the nucleus through the online website.”

Lines 348-349. “We use an online website to speculate the localization of the protein http://www.csbio.sjtu.edu.cn/bioinf/Cell-PLoc-2/.”

Point 9: Figure 2, images without scale bars

Response 9: Thanks for your suggestion. Line 132. We have added the scale bars and redone the figure in the manuscript, and now the nucleus spot of GsGIS3-GFP is visible.

Point 10: I am not sure about the significance in Figure 3G, please check it as the error bar at 8h is large and I can see that there is no significant difference between 0 and 8h.

Response 10: Thanks. As your suggestion, we removed the data with large errors and recalculated the significant differences between 0 and 8h. There was no significant difference between 0 and 8h. We have revised the mistake in the figure.

Point 11: (-1) in the title of the Y-axis in figure 4 should be superscript.

Response 11: Thanks. Line191. We revised it in the figure 4.

Point 12: Total root length in figure 5B is not described in materials and methods and how it was calculated for all treatment?

Response 12: Thanks. Lines 419-420. We added the methods in the manuscript:

“The whole seedlings were used to measure the root indicators (n=20 seedings per group).”

Point 13: More details about the mode of action of Hematoxylin staining. Why it is shallower or deeper?

Response 13: Thanks for your suggestion. We added the details about hematoxylin staining. The sentences were as follow:

“hematoxylin could easily combine with Al in the root to form a purplish red complex. The degree of Al staining combined with hematoxylin and root tip was deep, which means that more Al was accumulated in the root tip, which was sensitive to Al toxicity.”

Point 14: I do not recommend using the word “novel” in the title.

Response 14: Thanks for your suggestion. We remove the word “novel”. The title was revised as follow:

Heterologous Expression of a Glycine soja C2H2 Zinc Finger Gene Improves Aluminum Tolerance in Arabidopsis

Reviewer 3 Report

In the title, keep any of the "C2H2 Zinc Finger Gene or GsGIS3"

Abstract

"in acid soil" - acidic soils will be better

"GsGIS3, cloned from wild soybean was functionally verified to Al tolerance in Arabidopsis" - Need improvements

"MDA" - need to introduce an abbreviation

The basis for the selection of this gene needs to be provided in the abstract.  At present form of abstract, it looks like authors have randomly picked one Zinc Finger protein from wild soybean to clone and characterize. 

Introduction

acid soils- acidic soil

"As might be expected, phytohormones have been involved in the Al-stress response" - need improvement 

Line 88-89 Soybean is one of the most important oil and protein crops in the world, providing approximately 28% of the edible oil and 67% of the protein sources for human beings. Please add reference and check for protein-I think it should be 40% protein

Line 90-92 Owing to most of wild soybean growing in poor environment which forms many excellent characters under the influence of long-term bad environment, therefore, wild soybean is an important resource for improving soybean. Reframe the sentence
Line 116 Bioinformatics analysis predicted that GsGIS3 protein existed a localization signal in nuclear. -Reframe the sentence

Results

Figure 2 is not convincing, the nucleus spot is too small.

Figure 5 - the total root length and SA dose do not seem to be improved in transgenic. This doubt the entier conclusion that the gene has a role in Al tolerance. 

Discussion

Discussion around the related DEGs is missing. Similarly, authors have not discussed the minimal improvement in Al tolerance in all four transgenic lines. 

Author Response

Dear editors and reviewers,

We earnestly appreciate time and effort taken by the Editors and Reviewers to improve the content and clarity of our manuscript (ID ijms-699410). We have studied the comments carefully and have made corrections which we hope meet with your approval. We have submitted track-changed versions of manuscript. The main corrections in the manuscript and the response to each of the reviewer’s comments are detailed below. Thank you very much for your thoughtful comments and suggestions. We look forward to hearing from you regarding the suitability of the manuscript for publication in International Journal of Molecular Sciences.

Best regards,

Xiuxiang Zhang, PhD

Email: xiuxiangzh@scau.edu.cn

Response to Reviewer Comments

Point 1: In the title, keep any of the "C2H2 Zinc Finger Gene or GsGIS3"

Response 1: Thanks for your suggestion. We delete “GsGIS3” from the title. The title was revised as follow:

“Heterologous Expression of a Glycine soja C2H2 Zinc Finger Gene Improves Aluminum Tolerance in Arabidopsis”

Point 2: Abstract

"in acid soil" - acidic soils will be better

Response 2: Thanks. Line13, we have revised the sentence into

“Aluminum (Al) toxicity limits the plant growth and has a major impact on the agricultural productivity in acidic soils.”

Point 3:"GsGIS3, cloned from wild soybean was functionally verified to Al tolerance in Arabidopsis" - Need improvements

Response 3: Thanks. Lines 16-18 As your suggestion, the sentence has been rewritten as follow:

“In this study, GsGIS3 was isolated from Al-tolerant wild soybean gene expression profiles to be functionally characterized in Arabidopsis.”

Point 4:"MDA" - need to introduce an abbreviation

Response 4: Thanks. We added the explanation for “MDA”. The corresponding changes were as follows:

Lines 21-22, “higher proline and lower Malondialdehyde (MDA)…”

Line 59, “MDA content is an important parameter to…”

Point 5: The basis for the selection of this gene needs to be provided in the abstract. At present form of abstract, it looks like authors have randomly picked one Zinc Finger protein from wild soybean to clone and characterize. 

Response 5: Thanks for your suggestion. Lines 16-18,

we add the basis for this gene in the abstract, and the sentence was revised as follow:

“In this study, GsGIS3 was isolated from Al-tolerant wild soybean gene expression profiles to be functionally characterized in Arabidopsis.”

The gene was come from the basis of Al-tolerant wild soybean gene expression profiles (unpublished data), we mentioned the basis of this gene in Section 2.1, and we provided the part of data in the supplementary materials “Gene chip data analysis (root and shoot)”.

Point 6: Introduction

acid soils- acidic soil

Response 6: Thanks. Line37. Revised.

Point 7: "As might be expected, phytohormones have been involved in the Al-stress response" - need improvement 

Response 7: Lines 51-52. As your suggestion, the sentence has been revised as follow:

Thus, it can be seen that some phytohormones are involved in the Al-stress response.”

Point 8: Line 88-89 Soybean is one of the most important oil and protein crops in the world, providing approximately 28% of the edible oil and 67% of the protein sources for human beings. Please add reference and check for protein-I think it should be 40% protein

Response 8: Lines 90-91. Thanks. We added the reference in the manuscript and checked the protein content, the protein content of soybean is about 40%. Soybean provides approximately 67% of the plant protein for human beings. The sentence has been rewritten as follow:

“Soybean is one of the most important oil and protein crops in the world, providing approximately 28% of the edible oil and 67% of the plant protein sources for human beings [38].”

“Goldberg, B.; Stacey, G. Genetics and Genomics of Soybean; 2008.”

Point 9: Line 90-92 Owing to most of wild soybean growing in poor environment which forms many excellent characters under the influence of long-term bad environment, therefore, wild soybean is an important resource for improving soybean. Reframe the sentence

Response 9: Lines 92-97. As your suggestion, we have rewritten the sentence as follow:

“In the process of domestication and improvement of soybeans, the genetic diversity of soybeans has declined sharply. The annual wild soybean has a large amount of genetic diversity and preserved the excellent qualities lost in soybean. Therefore, Via our previous work, we constructed an Al-tolerant wild soybean gene expression profiles using the collected wild soybean resource in order to study the mechanism of resistance to acid aluminum stress.”

Point 10: Line 116 Bioinformatics analysis predicted that GsGIS3 protein existed a localization signal in nuclear. -Reframe the sentence

Response 10: Thanks. We revised this sentence and added part of methods in the manuscript. The sentences were revised as follow:

Lines 122-123. “We predicted that GsGIS3 protein was localized in the nucleus through the online website.”

Lines 348-349. “We use an online website to speculate the localization of the protein http://www.csbio.sjtu.edu.cn/bioinf/Cell-PLoc-2/.”

Point 11: Results

Figure 2 is not convincing, the nucleus spot is too small.

Response 11: Thanks for your suggestion. Line 132. We have redone the figure in the manuscript, and now the nucleus spot of GsGIS3-GFP is visible.

Point 12: Figure 5 - the total root length and SA dose do not seem to be improved in transgenic. This doubt the entier conclusion that the gene has a role in Al tolerance. 

Response 12: Thanks. We added some statistical data to make the conclusion more accurate. The added sentences were as follow:

Lines 191-197. “Under the control growth conditions, no obvious difference was found between wild type and transgenic Arabidopsis. The total root lengths of transgenic lines and WT-type were about 130 cm long, and the SA of that were about 12 cm2. However, under AlCl3 treatment, all observed plants performed delayed growth and depressed root elongation, the total root length of the WT-type was 29 cm long, while the total root lengths of GsGIS3 transgenic lines were ranged from 43 cm to 47 cm long. The SA of the WT-type was 2.2 cm2, while the SA of GsGIS3 transgenic lines were ranged from3.9 cm2 to 4.4 cm2.”

Point 13: Discussion

Discussion around the related DEGs is missing. Similarly, authors have not discussed the minimal improvement in Al tolerance in all four transgenic lines.

Response 13: Thanks for your suggestion. The related genes were obtained by the previous studies and references. We tried to investigate the potential transcription mechanism by the expression level of these genes, and we performed the experiments through one of the transgenic lines. The discussion about the related genes were as follow:

Lines 317-322.

“The increased transcript level of ALMT1 and ALS3 showed an improved Al tolerance capability in GsGIS3-over-expressing lines (Figure 7A, B), which was consistent with VuSTOP1 [44], GmSTOP1 [37] and WRKY46 [50]. ALMT1 which promote malate secretion, is related to the most Al tolerance phenotype in Arabidopsis, previous reports indicated that transcription factor WRKY46 could bind the promoter of ALMT1 to modulate Al stress tolerance [50], suggesting that GsGIS3 may function as a regulator of ALMT1 to improve Al tolerance.”

Lines 329-333。

“In our study, the transcription level of GA2OX1 was upregulated by GsGIS3, whereas the expression of GA3OX1 was reduced in transgenic Arabidopsis in Al stress (Figure 7C, D). Previous studies revealed that plants could improve the resistance to stress through reducing the accumulation of active gibberellin in plants [56], usually accompanied with the upregulation of GA2OX1 and the downregulation of GA3OX1 [57]”

Lines 338-342’

“Research above showed that the expression of GA3OX1 increased in the transgenic plant, while the expressions of GA2OXs decreased [59]. Furthermore, GID1 is the receptor of the active Gibberellin, the reduction of GID1 can also result in the decrease of active Gibberellin [60]. In our study, the expression of AtGID1 was reduced in GsGIS3 transgenic plants.”

Round 2

Reviewer 1 Report

The authors have addressed all the comments.

Author Response

Dear  reviewer,

We earnestly appreciate your time and effort taken to improve the content and clarity of our manuscript (ID ijms-699410). Thank you very much for your thoughtful comments and suggestions again. 

Best regards,

Xiuxiang Zhang, PhD

Email: xiuxiangzh@scau.edu.cn

Reviewer 2 Report

The authors responded positively to some of my comments and did language editing, however, the manuscript still needs extensive revision to address poor sentence structure to improve the quality of the manuscript. The discussion section needs to be improved and the authors should re-evaluate their conclusions.

Reviewer 3 Report

Authors have made significant improvement in the MS

Authors need to take proper care about English grammar and spelling mistakes

Author Response

Dear reviewer,

We earnestly appreciate your time and effort taken to improve the content and clarity of our manuscript (ID ijms-699410). We have our manuscript undergone extensive English editing by the MDPI English editing service and submitted track-changed versions of manuscript. The English grammar and spelling mistakes have been revised in the manuscript. Thank you very much for your thoughtful comments and suggestions again.

Best regards,

Xiuxiang Zhang, PhD

Email: xiuxiangzh@scau.edu.cn